# Respiratory Failure in Patients with Intracerebral Hemorrhage and Intraventricular Hemorrhage Extension: A Retrospective Study

**DOI:** 10.3390/healthcare13151876

**Published:** 2025-07-31

**Authors:** Min Cheol Chang, Michael Y. Lee, Sang Gyu Kwak, Ah Young Lee

**Affiliations:** 1Department of Physical Medicine and Rehabilitation, College of Medicine, Yeungnam University, Daegu 42415, Republic of Korea; wheel633@gmail.com; 2H. Ben Taub Department of Physical Medicine and Rehabilitation, Baylor College of Medicine, Houston, TX 77030, USA; Michael.Lee@bcm.edu; 3Department of Medical Statistics, College of Medicine, Catholic University of Daegu, Daegu 42472, Republic of Korea; sgkwak@cu.ac.kr; 4Department of Physical Medicine and Rehabilitation, Daegu Veterans Hospital, Daegu 42835, Republic of Korea

**Keywords:** intracerebral hemorrhage, intraventricular hemorrhage, respiratory failure, prediction, stroke

## Abstract

**Background/Objectives:** This study aimed to identify the risk factors for respiratory failure in patients with intracerebral hemorrhage (ICH) accompanied by intraventricular hemorrhage (IVH) extension. **Methods**: We retrospectively included 208 patients with ICH accompanied by IVH extension. Respiratory failure was defined as carbon dioxide levels > 45 mmHg with a pH < 7.35 in arterial blood gas analysis (ABGA) or the application of a ventilator due to respiratory dysfunction. We measured the severity of IVH extension using the Graeb scale, and ICH volume was assessed for each patient. **Results**: Of the 208 included patients, 83 had respiratory failure. There were no significant differences in age, sex ratio, or Graeb scale score between patients with and without respiratory failure (*p* > 0.05). However, ICH volume was significantly larger in patients with respiratory failure (42.0 ± 42.5 mL) than in those without (26.4 ± 25.7 mL) (*p* = 0.003). In the receiver operating characteristic (ROC) curve analysis, the area under the ROC curve for ICH volume predicting respiratory failure was 0.612. The optimal threshold for detecting respiration failure in patients with ICH and IVH dilatation, based on the Youden index, was >63.2 mL, with a sensitivity of 30.12% and a specificity of 89.60%. Approximately 40% of patients experienced respiratory failure following ICH accompanied by IVH extension. **Conclusions**: A large ICH volume was associated with the occurrence of respiratory failure. Therefore, caution is required in patients with an ICH volume > 63.2 mL.

## 1. Introduction

Non-traumatic spontaneous intracerebral hemorrhage (ICH) is a major public health issue with an annual incidence of 10–30 cases per 100,000 individuals, accounting for 10–15% of all strokes in adults [1,2]. ICH causes disturbances in various activities, including motor, sensory, cognitive, language, visual, eating, and respiratory functions [3,4,5]. Intraventricular hemorrhage (IVH) is initially present in approximately one-quarter of patients with ICH or can occur as a subsequent extension of the ICH component [6]. ICH and IVH extension increase intracranial pressure, which can result in brainstem compression [7]. Among the symptoms caused by brainstem compression from ICH and IVH extension, respiratory dysfunction is one of the most devastating, potentially leading to patient death [4,8].

Brainstem compression can affect the central control of the respiratory drive and the respiratory mechanics of inspiration and expiration [9]. The central control of breathing involves the complex integration of afferent signals, generation of a pattern of inspiratory and expiratory impulses, and transmission of efferent signals to respiratory muscles through descending tracts [10]. A compressed brainstem can alter the sensitivity and response to input from peripheral and central chemoreceptors, creating changes in the rate and rhythm of respiration [9,10]. Considering these mechanisms, it is predicted that patients with ICH accompanied by IVH extension might have a high possibility of respiratory failure.

Identifying which patients with ICH accompanied by IVH extension are at high risk of respiratory failure is crucial for the appropriate management and prevention of the devastating consequences of respiratory dysfunction. However, factors related to the development of respiratory failure in these patients have not been studied.

In the current study, we investigated the risk factors for the occurrence of respiratory failure in patients with ICH accompanied by IVH extension.

## 2. Materials and Methods

### 2.1. Subjects

This study was conducted retrospectively. Data on all patients admitted to the Department of Neurosurgery of our university between January 2006 and December 2023 were initially reviewed. The following criteria were then applied to identify eligible cases: (1) presence of non-traumatic spontaneous ICH accompanied by IVH extension, (2) first-ever stroke, (3) availability of arterial blood gas analysis (ABGA) data without oxygen supplementation, (4) patients whose ventilator use due to respiratory dysfunction or nonuse can be assessed on the basis of electrical health records, and (5) absence of serious medical complications that could affect respiratory function, such as pneumonia, pulmonary edema, or cardiac problems. All the included patients were initially evaluated by neurosurgeons upon admission. The diagnosis of ICH and IVH extension was confirmed by brain CT scan. ICH and IVH extension was defined as the presence of ICH identified on brain CT imaging with visible blood extending into the ventricular system.

The study protocol was approved by the Institutional Research Board of Yeungnam University Hospital, and the requirement for informed consent was waived because of the retrospective nature of this study.

### 2.2. An Evaluation of the Development of Respiratory Failure

We defined the presence of respiratory failure as carbon dioxide levels > 45 mmHg with a pH < 7.35 in ABGA or the application of a ventilator due to respiratory dysfunction [11]. Patients who had respiratory failure were classified into the RF group, and those who did not were classified into the non-RF group.

### 2.3. Severity of IVH Extension and Estimation of ICH Volume

To evaluate the severity of IVH extension and calculate ICH volume, initial brain computed tomography (CT) scans were used. The scan with the largest combined size of ICH and IVH was selected. IVH severity was graded using the Graeb scale on the CT images [12]. The Graeb scale is a semi-quantitative assessment tool used to score the extent of IVH, with a total range of 0 to 12 points. Higher scores indicate a greater volume of IVH. Each lateral ventricle could receive a score of up to 4 if it was enlarged and filled with blood, while the third and fourth ventricles could be assigned up to 2 points if they were both expanded and filled with blood.

ICH volume was calculated manually using the ellipsoid formula, based on measurements taken from CT scans [13]. The maximal diameter of the hematoma (A), its widest perpendicular diameter (B), and its vertical height (C) were measured in centimeters. The calculated volume was obtained using the following formula:volume = 4/3 × π × (A/2) × (B/2) × (C/2).

The ICH volume and Graeb scale score were measured by a single physician who was blinded to the patients’ respiratory failure status.

### 2.4. Statistical Analysis

Data were analyzed using Statistical Package for Social Sciences (SPSS ver. 22.0, Chicago, IL, USA). We assessed the normality of continuous variables using the Shapiro–Wilk test. All continuous variables, including age, Graeb scale score, and ICH volume, were found to follow a normal distribution. Potential confounding variables such as age, sex, ICH volume, and Graeb scale score were identified based on clinical relevance and the previous literature [14,15]. Age, sex, ICH volume, and Graeb scale score were compared between the RF and non-RF groups using the independent *t*-test and chi-squared test. In addition, we performed a receiver operating characteristic (ROC) analysis to assess the sensitivity, specificity, and cutoff value of clinical factors for predicting the development of respiratory failure in patients with ICH accompanied by IVH extension. The acceptable statistical significance level was set at *p* < 0.05.

## 3. Results

A total of 235 patients were recruited for this study. However, 27 patients were excluded due to the following: a previous history of stroke (seven patients), pneumonia (twelve patients), heart problems (three patients), or a lack of available data or records (five patients). Therefore, 208 patients were finally included in this study (Table 1). Among these, 83 were classified into the RF group and 125 into the non-RF group (Table 1). Patients with missing data were excluded from the analysis, resulting in the exclusion of five patients. For the 208 included patients, there were no missing data for variables such as age, sex, Graeb scale score, and ICH volume.

In the intergroup comparison between the RF and non-RF groups, there was no significant difference in age, sex ratio, and Graeb scale score (Table 1, *p* > 0.05). However, the ICH volume was significantly larger in the RF group compared to the non-RF group (Table 1, *p* < 0.05).

In the ROC curve analysis, the area under the ROC curve for ICH in predicting respiratory failure was 0.612 (95% confidence interval = 0.542–0.678; *p* = 0.006) (Figure 1). The optimal ICH volume threshold for predicting respiratory failure, based on the Youden index, was >63.2 mL, with a sensitivity of 30.12% and a specificity of 89.60%.

## 4. Discussion

In the current study, we observed that approximately 40% (83 out of 208) of the included patients with ICH accompanied by IVH extension experienced respiratory failure. Larger volumes of ICH were associated with an increased occurrence of respiratory failure; however, age, sex, and the severity of IVH extension did not affect occurrence.

Considering our results, despite the extensive extension of IVH following ICH, it appears that IVH does not mechanically affect the structures related to respiratory function in the brainstem. We hypothesize that the volume of IVH extension may not be large enough to compress the brainstem. In contrast, our findings suggest that patients with larger ICH volumes have a higher likelihood of experiencing respiratory failure. The association between larger ICH volumes and respiratory failure may be explained by the compression of the brainstem, which could lead to deterioration in the central control of the respiratory drive, mechanics of respiration, or transmission of respiratory signals. We found that approximately 40% of the patients with ICH accompanied by IVH extension experienced respiratory failure. This suggests that when ICH develops along with IVH, the ICH volume is often large enough to compress the brainstem. Clinicians should consider the possibility of the occurrence of respiratory failure and check the respiratory function in patients with ICH accompanied by IVH extension. In addition, our results showed that approximately 90% of the patients with an ICH volume < 63.2 mL did not experience respiratory failure, whereas approximately 30% of the patients with an ICH volume ≥ 63.2 mL had respiratory failure. Therefore, when the volume of ICH is >63.2 mL in patients with ICH accompanied by IVH extension, clinicians should pay particular attention to the development of respiratory failure.

Our proposed cutoff for ICH volume ≥ 63.2 mL demonstrated a low sensitivity of 30%, indicating that it would fail to identify 70% of patients who develop respiratory failure. While the high specificity of about 90% suggests that patients with an ICH below this cutoff are unlikely to develop respiratory failure, the low sensitivity limits its utility as a screening tool to rule out high risk. Therefore, clinicians should interpret this cutoff with caution, recognizing that although a larger ICH volume increases the likelihood of respiratory failure, many patients with smaller ICH volumes may still be at risk. Also, the AUC for ICH volume was statistically significant at 0.612, which indicates only weak discriminatory ability. This suggests that ICH volume alone can be a poor predictor of respiratory failure and that other clinical factors should be considered to improve prediction accuracy.

In addition, pulmonary complications such as pulmonary edema and pneumonia are common in patients with ICH and can also contribute to the development of respiratory failure [16]. In our study, patients with pulmonary complications such as pneumonia and pulmonary edema were excluded. Therefore, we believe that respiratory failure in the recruited patients was primarily caused by disturbances in the respiratory drive due to brainstem compression.

Regarding respiratory dysfunction after stroke, only a few studies have been conducted [4,17]. Previously, Rochester et al. reported that over 60% of patients with stroke develop respiratory dysfunction [4], and Cadilhac et al. found that over 80% of patients who had respiratory dysfunction at an early stage of stroke had continued deficits even 3 years after the stroke [17]. However, previous studies did not consider the severity of respiratory dysfunction or evaluate the risk factors associated with its development. Unlike previous studies, we included only patients experiencing significant respiratory dysfunction (carbon dioxide levels > 45 mmHg with a pH < 7.35 on ABGA or the application of a ventilator due to respiratory dysfunction) and observed that ICH volume is a critical factor in determining the occurrence of respiratory deterioration.

As for the development of respiratory failure after stroke, to our knowledge, only one study was published as a conference abstract [18]. Kim et al. recruited 220 patients with acute stroke [17]. Of these, 40 patients (18%) showed levels of carbon dioxide > 45 mmHg and oxygen < 60 mmHg on ABGA. The higher incidence of respiratory failure observed in our study compared to Kim et al.’s study may be attributed to the differences in patient populations. Kim et al.’s study included not only ICH patients but also cerebral infarct patients, and the stroke size and severity would have been different from those in our study. In Kim et al.’s conference abstract [18], the inclusion criteria and characters of included patients were not presented. Furthermore, risk factors for the occurrence of respiratory failure were not evaluated.

Our study is the first to report the occurrence rate of respiratory failure in patients with ICH accompanied by IVH extension and to evaluate the risk factors for developing respiratory failure. However, our study had several limitations. First, it was conducted retrospectively. Second, only patients with a specific type of stroke were included. Third, we did not calculate an appropriate sample size for our study, and the analysis was based on a convenience sample of 208 patients. This may have limited the statistical power to detect small effects. For example, although the Graeb scale score did not show a significant association with respiratory failure in our analysis, it is possible that this non-significant finding was due to insufficient sample size rather than a true lack of association. Fourth, we did not investigate various clinical factors such as comorbidities or ICH location, which may have affected the occurrence of respiratory failure. Fifth, our study may have selection bias because patients with very early, catastrophic hemorrhage who had immediate withdrawal of care might have been excluded, potentially leading to an underestimation of the true incidence of respiratory failure. Lastly, the CT scans used to measure ICH volume were performed at slightly different time points after the onset. Future studies addressing these limitations are warranted.

## 5. Conclusions

In the current study, we found that larger ICH volume was statistically associated with the occurrence of respiratory failure in patients with ICH accompanied by IVH extension. However, the discriminatory ability of ICH volume was weak, and the proposed cutoff of 63.2 mL showed low sensitivity despite high specificity. Therefore, while caution is warranted in patients with a large ICH volume, ICH volume alone is a poor predictor of respiratory failure. The proposed cutoff should be considered exploratory and requires further valuation in larger, prospective studies.

## Figures and Tables

**Figure 1 healthcare-13-01876-f001:**
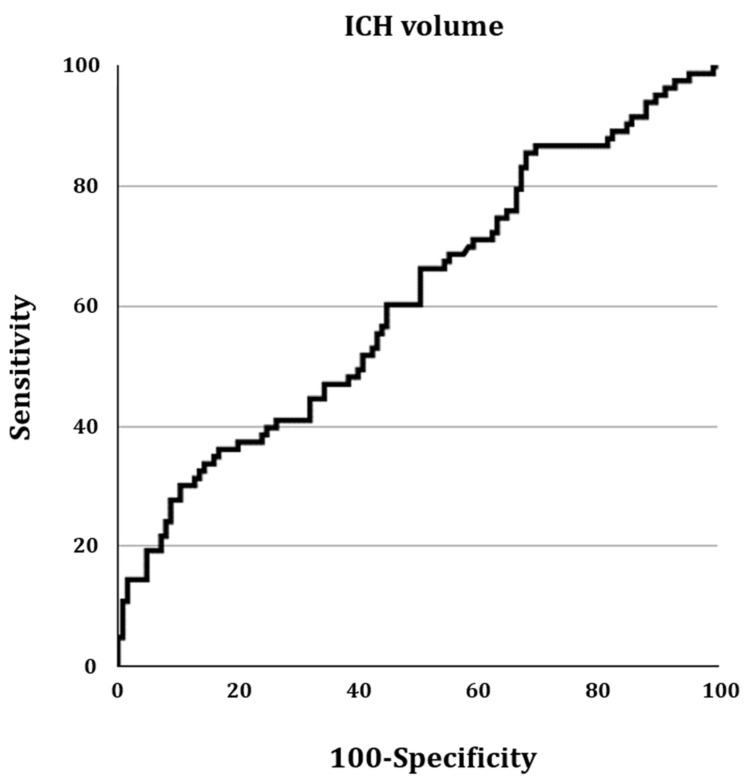
The receiver operating characteristic curve of the intracerebral hemorrhage (ICH) volume in patients with ICH accompanied by intraventricular hemorrhage extension.

**Table 1 healthcare-13-01876-t001:** Demographic data, Graeb scale score, and intracerebral hemorrhage in relation to the development of respiratory failure.

	Total (n = 208)	RF Group (n = 83)	Non-RF Group (n =125)	*p*-Value	95% CI (Lower Limit, Upper Limit)
Age (years), mean ± SD	58.4 ± 13.0	58.2 ± 11.5	58.8 ± 13.9	0.782	−3.123, 4.145
Sex (M:F)	110:98	43:40	67:58	0.800	
Graeb scale score, mean ± SD	6.3 ± 3.0	6.6 ± 2.9	6.2 ± 3.0	0.324	−1.253, 0.417
ICH volume (mL), mean ± SD	32.6 ± 34.2	42.0 ± 42.5	26.4 ± 25.7	**0.003**	−25.905, −5.331

Abbreviations: RF, respiratory failure; SD, standard deviation; CI, confidence interval; ICH, intracerebral hemorrhage. Bold indicates *p* < 0.05.

## Data Availability

Data are contained within the article.

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
