# Peer review of "Respiratory Failure in Patients with Intracerebral Hemorrhage and Intraventricular Hemorrhage Extension: A Retrospective Study"

_healthcare, 2025, doi:10.3390/healthcare13151876_

Round 1

Reviewer 1 Report

Comments and Suggestions for Authors

Review Comments

The study titled “Respiratory Failure in Patients with Intracerebral Hemorrhage and Intraventricular Hemorrhage Extension: A Retrospective Study” evaluated the occurrence of respiratory failure in patients with intracerebral hemorrhage (ICH) accompanied by intraventricular hemorrhage (IVH) extension. In this retrospective study, authors demonstrated that larger ICH volume is correlated to respiratory failure in patients with ICH with IVH extension. This study provides valuable insight into the clinical management of ICH patients with IVH extension, particularly in monitoring for respiratory complications., however there are minor suggestions listed below:

Minor Comments:

  1. The authors report that 30% of patients with an ICH volume ≥ 63.2 mL developed respiratory failure and emphasize the need for clinical vigilance in this subgroup. It would strengthen the manuscript if the authors could provide any relevant references from literature to support this threshold as predictive or indicative of severe outcomes, particularly in relation to respiratory failure
  2. Although the study is presented as a short communication, inclusion of a summarized table of patient demographics would represent a clearer clinical profile of the study cohort.
  3. While the manuscript states that no sex-based differences were observed, it is important to clarify the distribution of respiratory failure cases between male and female patients with ICH and IVH extension. Providing these subgroup numbers would enhance the transparency and interpretability of the sex-related findings.

Author Response

Review Comments

The study titled “Respiratory Failure in Patients with Intracerebral Hemorrhage and Intraventricular Hemorrhage Extension: A Retrospective Study” evaluated the occurrence of respiratory failure in patients with intracerebral hemorrhage (ICH) accompanied by intraventricular hemorrhage (IVH) extension. In this retrospective study, authors demonstrated that larger ICH volume is correlated to respiratory failure in patients with ICH with IVH extension. This study provides valuable insight into the clinical management of ICH patients with IVH extension, particularly in monitoring for respiratory complications., however there are minor suggestions listed below:

Answer: I appreciate your kind comments.

Minor Comments:

Comments 1: The authors report that 30% of patients with an ICH volume ≥ 63.2 mL developed respiratory failure and emphasize the need for clinical vigilance in this subgroup. It would strengthen the manuscript if the authors could provide any relevant references from literature to support this threshold as predictive or indicative of severe outcomes, particularly in relation to respiratory failure.

Answer 1: I appreciate your comments. Because this is first study to find ICH volume that cause respiratory failure in ICH, we cannot provide any reference associated to our results.

Comments 2: Although the study is presented as a short communication, inclusion of a summarized table of patient demographics would represent a clearer clinical profile of the study cohort.

Answer 2: I appreciate your comment. I agree with your comment. However, we did not investigate various clinical factors such as comorbidities or ICH location, which can affect the occurrence of respiratory failure. We added this matter to the main text as a limitation. (page number 5, line 201-203)

Comments 3: While the manuscript states that no sex-based differences were observed, it is important to clarify the distribution of respiratory failure cases between male and female patients with ICH and IVH extension. Providing these subgroup numbers would enhance the transparency and interpretability of the sex-related findings.

Answer 3: I appreciate your comment. We presented the ratio between male and female in Table 1. In RF group, the ratio (M:F) was 43:40, and in non-RF group, the ratio was 67:58. (page number 3, Table 1)

Reviewer 2 Report

Comments and Suggestions for Authors

Thank you for the opportunity to review your manuscript, "Respiratory Failure in Patients with Intracerebral Hemorrhage and Intraventricular Hemorrhage Extension: A Retrospective Study." This study addresses an important and clinically relevant question regarding the prediction of a devastating complication in a high-risk stroke population. The work is well-motivated, and the central finding that ICH volume is a more significant predictor of respiratory failure than IVH severity is an interesting contribution to the field.
The manuscript is clearly written and logically structured. Below are my detailed comments, which are intended to help strengthen the manuscript for publication.

  • Major Comments
  1. The main concern here is the lack of multivariable analysis, which is a key part of good analytical practice and is clearly outlined in the STROBE guidelines. The authors have used only basic, unadjusted tests like t-tests and chi-square tests, which means any potential confounding factors, like age or sex, haven't been accounted for. Even if some of these weren't significant in the bivariate analysis, they could still influence the relationship between ICH volume and respiratory failure. I'd strongly suggest including a multivariable logistic regression to address this, which would help provide more reliable, adjusted estimates and strengthen the study's conclusions.
  2. The authors noted the lack of a formal sample size calculation in the limitation section, but its impact should be addressed more clearly in the Methods or Discussion. With a convenience sample of 208 patients, the study may be underpowered to detect smaller effects, such as for the Graeb scale score. Please add a sentence in the discussion section acknowledging that the non-significant Graeb scale result could be due to limited power and that larger studies are needed to confirm its role.
  3. Were the individuals who calculated the ICH volume and Graeb scale scores blinded to the patient's outcomes (i.e., whether they were in the RF or non-RF group)? If not, this is a limitation that should be noted.
  4. Could there be systematic differences in the population from which cases and controls were drawn? For example, were patients with very early, catastrophic hemorrhages leading to immediate withdrawal of care excluded, potentially underestimating the true incidence of respiratory failure?
  5. The reported AUC of 0.612 is statistically significant but reflects weak discriminatory ability. This suggests that ICH volume alone is a poor predictor of respiratory failure. I recommend clearly stating this in the Discussion to provide a more balanced interpretation of its clinical relevance.
  6. The proposed cutoff of >63.2 mL has a very low sensitivity of 30.12%. This is a major practical limitation. This means that your proposed test would fail to identify nearly 70% of patients who develop respiratory failure. The high specificity (89.6%) is valuable for "ruling in" low risk, but the poor sensitivity severely limits its use as a screening tool to "rule out" high risk. The Discussion currently misinterprets this finding by stating, "About 30% of the patients with an ICH volume ≥ 63.2 mL had respiratory failure." This is incorrect. Sensitivity means that of all patients who had respiratory failure, only 30.12% had an ICH volume above the cutoff.
  7. The clinical implications of the 63.2 mL cutoff are overstated, given the weak AUC and low sensitivity. While caution is reasonable for patients above this volume, the overall message needs to be more measured. I recommend adjusting the conclusions to reflect that ICH volume is a statistically significant but weak predictor, and the proposed cutoff is exploratory and requires further validation before clinical application.
  8. The Discussion includes language that implies causation, such as "Large ICH volumes appear to compress the brainstem…" which is not appropriate for a retrospective, observational study. Please rephrase these statements to reflect an association rather than causation; for example, "The association between larger ICH volumes and respiratory failure may be explained by brainstem compression…".
  9. The section titled "Severity of IVH extension and estimation of ICH volume" appears to be directly copied from your previously published paper (DOI: https://doi.org/10.3340/jkns.2024.0192). This raises a serious concern regarding self-plagiarism. Please address this issue by either appropriately paraphrasing the content with proper citation or removing the duplicated text.
  • Minor Comments

Abstract

  1. Please add the subheading "Results" before this sentence: "Of the 208 included patients, 83 had respiratory failure, while the other 125 did not."
  2. Please consider adding more numerical data to the abstract results.

Methods

  1. There's a contradiction regarding which CT scan was used for ICH volume calculation. One part of the text states that the scan with the largest hemorrhage was selected, while another mentions using the initial CT. Please clarify this point. Using the scan with the largest hemorrhage is reasonable but introduces variability if the timing differs across patients; this should be clearly stated along with the rationale.
  2. The Results mention that 5 patients were excluded due to missing data, but the Methods section does not explain how missing data were handled. Please clarify this in the Methods, specifying that case exclusion was used. Also, kindly confirm whether there were any missing data for key variables such as age, sex, Graeb score, and ICH volume within the final cohort of 208 patients.
  3. Line 81: Define the abbreviation (RF)

Results

  1. The 95% CI provided for the "Sex (M: F)" comparison is unconventional for a chi-square test. It appears to be a CI for an odds ratio, but the odds ratio itself is not provided. Please clarify this or remove the CI for this categorical variable comparison.

Discussion & Conclusion

  1. The Conclusion states that "particular caution is needed in patients with ICH volume ≥ 60 mL." The value derived in the Results was 63.2 mL. Please maintain consistency.

General comment

  1. The name "Myung Sub Yi" appears in the Author Contributions section but not in the main author list at the beginning of the manuscript. This is a significant discrepancy that must be corrected.
  2. The abbreviations section is not comprehensive and misses several abbreviations.

Reviewer 3 Report

Comments and Suggestions for Authors

Chang et al. conducted a retrospective study on respiratory failure in patients with intracerebral hemorrhage (ICH) and associated intraventricular hemorrhage (IVH) extension.

The authors state that no funding was received for this research. Could they clarify how this was determined and whether there were any institutional or departmental resources involved?

The selection criteria need to be described in greater detail. How were patients initially selected? Were all potential cases reviewed and then screened using the listed criteria, or were the criteria applied at the outset to identify eligible cases?

How was the etiology of the hemorrhage assessed? Were multiple etiologies considered and included? Please explain the process and basis for final diagnosis of the cause of bleeding.

Were neurologists and neurosurgeons involved in the diagnostic process? If so, please elaborate on how the diagnosis was established and whether standardized diagnostic protocols were followed.

The authors should be specific in listing the factors that contributed to respiratory failure in this patient population. Which cases were excluded from the study, and for what reasons? Avoid generalized statements, as they may impact the study’s validity.

How was the volume of bleeding calculated? The current manuscript mentions manual estimation—please confirm and specify the method used.

Please clearly define the criteria used to classify ICH with IVH extension.

How were the variables distributed? Indicate which variables followed a normal distribution. In the statistical analysis section, specify whether any statistical software was used to assess the ICH and to check for normality of the data.

How were confounding variables identified and accounted for in the analysis?

The manuscript should be revised to conform to the journal’s “Instructions for Authors.” Currently, the formatting of references, in-text citations, and figure legends is inconsistent with these guidelines.

The final paragraph of the discussion lacks a section on the limitations and potential weaknesses of the study. This should be added for transparency.

Please clarify how this study adds novel insights compared to existing literature. What distinguishes this study from previous similar investigations?

Why were comparative statistical methods or regression analyses not performed? Why was ROC analysis conducted directly instead? Was this analytical approach defined in the original study design?

Round 2

Reviewer 3 Report

Comments and Suggestions for Authors

I do not have further comments.

Author Response

The authors examine a population with spontaneous ICH with IVH. They show a correlation between ICH volume and respiratory failure but no correlation with IVH volume. Others have shown that ICH volume is associated with increased morbidity and mortality. The current finding that increased ICH volume is associated with RF is not unexpected or novel. It would be of greater interest if the authors examined whether ICH + IVH has a higher risk for RF compared to similar volumes of ICH without IVH. Lastly, pulmonary edema and pneumonia are the most common pulmonary complications in ICH. It would be helpful if the authors further define the etiologies of RF.

Answer: I appreciate your comments. The finding that increased ICH volume is associated with respiratory failure can be expected. However, this is first to report the occurrence rate of respiratory failure in patients with ICH accompanied by IVH extension and to evaluate the risk factors for developing respiratory failure. Surely, the results can be expected, but this is the first actual study. Also, I fully agree that the examination whether ICH + IVH has a higher risk for respiratory failure compared to similar volumes of ICH without IVH. It is good topic. However, in this stage, it is hard to recruit the patients with adjusting ICH volume. I think that it is a good topic for the future studies. I hope that you can understand our situation.

Regarding the last comment, pulmonary edema and pneumonia can be one of the most common pulmonary complications in ICH. I agree with your comments. In our study, patients with pulmonary complications such as pneumonia and pulmonary edema were excluded. Therefore, we believe that respiratory failure in the recruited patients was primarily caused by disturbances in the respiratory drive due to brainstem compression. We clearly stated this matter to the discussion section.